# Preparation and In Vitro Release of Total Alkaloids from *Alstonia Scholaris* Leaves Loaded mPEG-PLA Microspheres

**DOI:** 10.3390/ma12091457

**Published:** 2019-05-06

**Authors:** Xiangyu Zheng, Hongli Li, Yi He, Mingwei Yuan, Meili Shen, Renyu Yang, Nianfeng Jiang, Minglong Yuan, Cui Yang

**Affiliations:** 1National and Local Joint Engineering Research Center for Green Preparation Technology of Biobased Materials, Yunnan Minzu University, Kunming 650500, China; zhengxiangyu1993@163.com (X.Z.); honglili@vip.163.com (H.L.); heyi_sichuan@163.com (Y.H.); yuanmingwei@163.com (M.Y.); yangrenyu1995@163.com (R.Y.); jnf1514@163.com (N.J.); 2National and Local Joint Engineering Laboratory for Synthesis Technology of High Performance Polymer, Jilin University, Changchun 130012, China; sml199211@163.com

**Keywords:** total alkaloids from *Alstonia scholaris* leaves, mPEG-PLA, microspheres, drug release, biocompatibility

## Abstract

Total alkaloids of *Alstonia scholaris* leaves (ASAs) are extracted from the lamp leaves, which have positive anti-inflammatory activity and remarkable effects in treating bronchitis. Due to its short half-life, we used a degradable mPEG-PLA to physically encapsulate the total alkali of the lamp stage, and prepared a sustained-release microsphere by double-emulsion method. The ASAs-loaded mPEG_10000_-PLA microspheres were screened for better performance by testing the morphology, average particle size, embedding rate and drug loading of different molecular weight mPEG-PLA microspheres, which can stably and continuously release for 15 days at 37 °C. The results of cytotoxicity and blood compatibility indicated that the drug-loaded microspheres have beneficial biocompatibility. Animal experiments showed that the drug-loaded microspheres had a beneficial anti-inflammatory effect. These results all indicated that mPEG-PLA is a controlled release carrier material suitable for ASAs.

## 1. Introduction

Respiratory diseases are common and frequently occurring diseases whose incidence has increased in recent years. Inhalation of dust and irritating gases are the main causes of respiratory diseases, which cause tremendous harm to human health [1]. *Alstonia scholaris* leaves, a traditional Chinese medicine from the Dai nationality, are mainly used to treat chronic respiratory diseases [2]. *A. scholaris* leaves are also used as a traditional medicine to treat respiratory diseases in India, Malaysia, the Philippines, and Thailand [2]. *A. scholaris* leaf extracts, especially alkaloids, are widely used to treat bronchitis and have significant efficacy [3,4,5,6,7]. Luo et al. studied *A. scholaris* leaves and found an alkaloid component that beneficial effects respiratory diseases. This component has a positive effect on bronchitis and post-cold infections [8,9]; however, its half-life is short, which limits its clinical application.

Research and development of sustained-release formulations have always been a focus in drug research and have gradually begun to solve the problem of short half-lives. In recent years, polymer nanoparticles have shown to be efficient drug carriers for encapsulating drugs [10]. At present, designing and selecting drug nanocomposites has mainly focused on amphiphilic block copolymers. Polyethylene glycol (PEG) is often used in the hydrophilic segment of the copolymers, while many homopolymers, copolymers and derivatives of degradable polymers, such as polylactic acid (PLA), polyglycolic acid (PGA) and polycaprolactone (PCL), are used in the hydrophobic segment [11,12,13,14]. Methoxy poly(ethylene glycol)-poly(lactide) copolymer (mPEG-PLA) is an amphiphilic polymer formed by grafting mPEG onto PLA. mPEG-PLA has excellent biocompatibility, a low molecular weight, and many hydroxyl groups and is nontoxic and widely used as a coating material in drug-delivery systems [15,16]. PEG blocks can improve the polymer’s hydrophilicity and flexibility, prevent protein adsorption and avoid recognition and phagocytosis by the reticuloendothelial system (RES). PEG encapsulates the drug to form the “core”, while the “shell” is formed by the outside hydrophobic segment, which constitutes the typical “core-shell” structure, with great advantages for drug release [17,18,19]. Therefore, in this work, total alkaloids from *A. scholaris* leaves (ASAs) were studied, and mPEG-PLA with different molecular weights was used as the carrier material. A series of drug-loaded microspheres were prepared by the water-oil-water (W/O/W) double-emulsion technique. Based on the morphology, particle size, encapsulation efficiency (EE), drug-loading efficiency (LE), and in vitro drug release of the prepared microspheres, the best drug-loaded materials were screened out, and then the biological properties of its drug-loaded microspheres such as cytotoxicity, blood compatibility and anti-inflammatory activity were studied.

## 2. Materials and Methods 

### 2.1. Materials

L-lactide was purchased from Purac Co., Ltd. (Shanghai, China). Dichloromethane (DCM), ethanol, tert-butyl methyl ether, and isopropanol were purchased from Tianjin Damao Chemical Reagent Co., Ltd. (Tianjin, China) and were all analytical grade. ASAs were purchased from Kunming Institute of Botany, Chinese Academy of Sciences, batch number 20180501 (Kunming, China). ASAs is a mixture of brown solid powder. Two of main active ingredients are Vallesamine and Picrinine, which were detected by HPLC with peak times of 43.133 min and 72.190 min. NaH_2_PO_4_, KH_2_PO_4_, and NaCl were purchased from Chengdu Chron Chemicals Co., Ltd. (Chengdu, China). Stannous octoate (95%, analytical grade) and polyvinyl alcohol (PVA; Mw = 75,000 Da and 88% alcoholysis degree) were purchased from Shanghai Jingchun Chemicals Co., Ltd. (Shanghai, China). mPEG (Mw 5000) was purchased from Shanghai Seebio Biotechnology Co., Ltd. (Shanghai, China). HL-7702 cells (normal human hepatocytes) were purchased from Procell Life Sciences and Technology Co., Ltd. (Wuhan, China). High glucose and 1640 medium were provided by HyClone Company. The 3-(4,5-dimethyl-2-thiazolyl)-2,5-diphenyl-2-H-tetrazolium bromide (MTT) was purchased from APExBIO (Houston, TX, USA). High-quality fetal bovine serum (FBS) was purchased from Shanghai ExCell Bio Co., Ltd. (Shanghai, China). Ethylenediaminetetraacetic acid (EDTA)-trypsin solution, cyan-streptomycin, and cisplatin were provided by Solarbio (Shanghai, China). Kunming mice, Sprague-Dawley (SD) rats, and rabbits were purchased from Kunming Medical University (Kunming, China). Dengtaiye Pian (DP) was purchased from Yunnan Datang Hanfang Pharmaceutical Co., Ltd. (Kunming, China). Aspirin was purchased from Bayer HealthCare Manufacturing Co., Ltd. (Beijing, China). 

### 2.2. Preparation of mPEG-PLA

mPEG-PLA was prepared by ring-opening polymerization. L-LA (18 g) and mPEG (2 g) were putted in reaction flask as raw materials. Protected by nitrogen, catalyzed by stannous octoate (0.02 g), the mPEG-PLA was polymerized at 130–160 °C for 24 h. Using recrystallization, the product was precipitated by methylene chloride and ethanol. Product was filtered and washed with hot water to remove the mPEG bulk polymer. The obtained product was dried with P_2_O_5_ for 48 h to obtain a white copolymer. Figure 1 shows the synthesis process.

### 2.3. Determination of Molecular Mass 

3 mg copolymer were dissolved in 1 mL chromatography grade THF and was filtered by using a 0.45 μm nylon 66 filter membrane. Then using gel permeation chromatography (GPC) which was purchased from Waters Inc. (Milford, MA, USA) to define the molecular weight and distribution of the compounds, and THF was used as the eluent. The system is equipped with a column (7.8 × 300 mm, Waters Styragel, Waters Inc., Milford, MA, USA), a Waters 515 pump and a Waters 2414 refractive index detector. When the column temperature of GPC is 40 °C, the flow rate is 1 mL/min, and the baseline is smooth. The filtrate was pulled into an injection needle, and the sample was slowly and uniformly injected into the sampler when the air was removed. All data were obtained under the same standard curve.

### 2.4. Preparing the Blank and ASAs-Loaded mPEG-PLA Microspheres

The ASAs-loaded mPEG-PLA microspheres were prepared using the W/O/W double-emulsion technique. First, 100 mg of ASAs were dissolved in 1 mL methanol solution, which was the internal water phase (W_1_). The 500-mg mPEG-PLA copolymer material was weighed and completely dissolved in 10 mL of methylene chloride as the oil phase (O). The external water phase (W_2_) was a 2% PVA solution. Next, the ASAs solution was injected into the oil phase in an ice bath at a constant rate. The mixture was emulsified at high speed (21,000 rpm) for 2 min to form the first emulsion. Similarly, the first emulsion was continuously dropped into 20 mL of 2% PVA (W_2_) in an ice bath and emulsified at high speed (21,000 rpm) for 2 min to form a double emulsion. The double emulsion was poured into 400 mL of 5% isopropanol solution and stirred at low speed at room temperature for 6 h. After the organic solvent had completely evaporated, the ASAs-loaded microsphere solution was obtained. Finally, the ASAs-loaded microsphere solution was centrifuged (6500 rpm, 10 min), and the supernatant was discarded. The microspheres were collected (white solid), washed with pure water 3 times, and lyophilized at −50 °C for 24 h. The obtained microspheres were dried at −20 °C. Blank microspheres were prepared by the same method.

### 2.5. Morphology, Particle size and Particle Size Distribution of the Microspheres

The morphological characteristics of the ASAs-loaded mPEG-PLA microspheres were observed using SEM (NOVA NANOSEM-450, FEI, Hillsboro, OR, USA). Conductive adhesive tape was applied to the surface of the loading platform. A few lyophilized microspheres (white powder) were evenly spread on a conductive adhesive, and the surface was sprayed with gold to observe. The particle size distribution of the microspheres was measured using a laser particle-size analyzer (Mastersizer 3500, Microtrac, Malvern, UK). A small portion of the microsphere suspension was diluted in a 15 mL centrifuge tube until the solution became nearly transparent. The particle size distribution of the microspheres was determined using a particle-size analyzer. 

### 2.6. Measuring the EE of the Microspheres

Twenty milligrams of lyophilized microspheres were weighed, and 500 μL of methylene chloride was added to completely dissolve them. Next, 2 mL of methanol solution was added. After centrifugation, the supernatant was filtered through a 0.45-μm microfiltration membrane. Next, 20 μL of filtered solution was injected via HPLC, and the area normalization method was used to calculate the EE% and LE% using the following equations.
$$\mathrm{LE}\%=\frac{\mathrm{weight}\;\mathrm{of}\;\mathrm{ASAs}\;\mathrm{in}\;\mathrm{microspheres}}{\mathrm{total}\;\mathrm{weight}\;\mathrm{of}\;\mathrm{microspheres}}\times 100\%$$
$$\mathrm{EE}\%=\frac{\mathrm{weight}\;\mathrm{of}\;\mathrm{encapsulated}\;\mathrm{drug}}{\mathrm{weight}\;\mathrm{of}\;\mathrm{initial}\;\mathrm{drug}\;\mathrm{loading}}\times 100\%$$

### 2.7. In Vitro Microsphere Release

Fifty milligrams of lyophilized microspheres were weighed and placed in a 15 mL centrifuge tube, then 5 mL of phosphate buffer saline (PBS) was added and shaken in a thermostatic oscillator at 37 °C. The tube was removed at set time intervals and centrifuged at 5000 rpm for 6 min. Next, 1 mL of the release solution was removed from the supernatant, an equal amount of fresh PBS was added to the tube, and the tube was placed back in the thermostatic oscillator and shaken at 37 °C. The ASAs content in the release solution was determined by HPLC, and the cumulative drug release amount was calculated using the following formula [20]. The amount (%) of the released ASAs-loaded microspheres was plotted against the release time (t, in days) to obtain the in vitro release curve of the microspheres.
$$\;Q=C_{n}\times V_{1}+V_{2}\sum C_{n-1}\;$$
Q: cumulative drug release (μg);C_n_: concentration of the release solution (μg/mL) removed at time t;V_1_: volume of the released medium (mL);V_2_: volume of the medium to be withdrawn each time (mL).

### 2.8. In Vitro Microsphere Degradation

Fifty milligrams of lyophilized microspheres were weighed and placed in a 15 mL centrifuge tube, and then, 5 mL of PBS was added and shaken at 37 °C. The tube was removed at set time intervals and centrifuged at 5000 rpm. The supernatant was collected to determine the pH changes in the microspheres. Buffer salts on the microsphere surface were washed with pure water and frozen at −50 °C for 24 h. The microspheres were weighed using an analytical balance, and the dry weight loss of the degraded microspheres was calculated. The molecular weight of 3 mg of the lyophilized microspheres was measured via GPC. The pH value, the microsphere weight and the relative molecular weight (Mn) were separately plotted against degradation time to obtain the in vitro degradation curve of the microspheres.

### 2.9. Blood Compatibility Testing of The Microspheres

All the trials were approved by the laboratory animals Ethics Committee of Yunnan Minzu University and were registered on the Kunming Science and Technology Bureau (SYXK (Yunnan) K2017-0001, 16 January 2017).

#### 2.9.1. Hemolysis Experiment

The samples were dissolved in deionized water (DI water), and the solutions were prepared at concentrations of 10.6 μg/mL, 1.06 mg/mL, and 0.106 mg/mL. Eight milliliters of fresh anticoagulated rabbit blood (blood mixed with sodium citrate at a volume ratio of 9:1) was diluted with 10 mL of 0.9% sodium chloride solution. Two hundred microliters of the sample was placed in a test tube, and 5 mL of 0.9% sodium chloride solution was added. The tube was then kept in a 37 °C water bath for 30 min. Next, 100 μL of diluted blood was added, and the mixture was gently mixed and incubated for 60 min. The positive control was treated with 5 mL DI water and 100 μL blood (D = 0.8 ± 0.3). The negative control was treated with 5 mL 0.9% NaCl solution and 100 μL blood. After centrifugation at 800 r/min for 5 min, the supernatant was transferred into a cuvette, and an ultraviolet (UV) spectrophotometer was used to measure the absorbance at 540 nm. Hemolysis of the sample was calculated per the formula below. The final sample concentrations were 40 μg/mL, 4 μg/mL, and 0.4 μg/mL. The material was considered hemolyzed when the hemolysis rate was above 5% [21,22,23,24,25,26,27].
$$\mathrm{Hemolysis}\;\mathrm{rate}\left(\%\right)\;=\;\frac{{\mathrm{Abs}}_{\mathrm{sample}}{-\;\mathrm{Abs}}_{\mathrm{negative}\;\mathrm{control}}}{{\mathrm{Abs}}_{\mathrm{positive}\;\mathrm{control}}{-\;\mathrm{Abs}}_{\mathrm{negative}\;\mathrm{control}}}\times 100\%$$

#### 2.9.2. Coagulation Experiment

The samples were dissolved in distilled water to prepare solutions with concentrations of 245.2 μg/mL, 24.52 μg/mL, and 2.452 mg/mL. The final concentrations were 40 μg/mL, 4 μg/mL, and 0.4 μg/mL. Next, 200 μL of the solution was placed in a 15-mL centrifuge tube and kept at 37 °C for 5 min. Fifty microliters of fresh anticoagulated rabbit blood was added to the samples, which were and kept at a constant temperature for 5 min. Ten microliters of aqueous calcium chloride solution (0.2 mL/L) was added to the blood sample, and the centrifuge tube was shaken to evenly mix the calcium chloride and blood and kept at a constant temperature for 5 min. The centrifuge tube was then removed, 12 mL of DI water was added to the solution, and the supernatant was collected. The blood was measured at 540 nm using a UV spectrophotometer. The optical absorbance (i.e., the optical density [OD] of the free hemoglobin remaining in the beaker containing 50 μL of whole blood treated with 12 mL of DI water) was used as a reference. The average of 5 measurements was taken. The sample’s anticoagulant activity was expressed as the relative absorbance [28,29]:
$$\mathrm{BCI}=\frac{I_{o}}{I_{w}}\times 100\%$$
I_o_: relative absorbance of the mixture of blood and calcium chloride after contact with the sample for a set period of time;I_w_: relative absorbance of blood mixed with a certain amount of DI water.

### 2.10. Cytotoxicity of The Microspheres

#### 2.10.1. Cell Culture

Normal human hepatocytes (HL-7702) were inoculated into culture flasks, and RPMI-1640 medium (containing 10% FBS, 100 U/mL penicillin, and 100 U/mL streptomycin) was added. Cells were cultured in a 5% CO_2_ incubator at 37 °C. Cells grew as monolayers adherent to glass and were passaged once every 3–5 days using 0.25% trypsin.

#### 2.10.2. MTT Assay of Sample Inhibition on Tumor Cell Proliferation 

HL-7702 cells (180 μL; 5 × 10^4^ cells/mL) in the logarithmic growth phase were inoculated in a 96-well plate, and 20 μL of the sample was added to each well after incubating overnight. Three concentration gradients were set with 3 wells per concentration. After a 48-h incubation, 20 μL of 5 mg/mL MTT was added to each well, and the cells were continuously cultured for 4 h. The culture medium was aspirated and discarded, and 150 μL dimethylsulfoxide (DMSO) was added to terminate the reaction. The plate was shaken for 15 min in a shaker. The OD value at 490 nm was measured in a microplate reader [30,31], and the inhibition rate was calculated [32]:
$$\mathrm{Cell}\;\mathrm{viability}=\frac{\mathrm{ODsample}}{\mathrm{ODnegative}\;\mathrm{control}}\times 100\%$$

### 2.11. Anti-Inflammatory Activity Testing of The Microspheres

To determine the influence of xylene-induced auricle swelling on the mice [33], 35 Kunming mice (19–22 g) were randomly divided into 7 groups by body weight and sex, with 5 mice per group. Except for the aspirin group, only one intragastric administration on the day of the experiment, the other groups were pre-administered once daily for 3 consecutive days. The control group was administered the same volume of 1% CMC-Na (carboxymethyl cellulose-sodium). The gavage volume was 20 mL/kg per group. Thirty minutes after the last gastric gavage, 0.05 mL of xylene was evenly applied on both sides of each mouse’s right ear, while the left ear was used as the control. The mice were sacrificed via cervical dislocation 1 h after inflammation. The same area of both ears was cut off using a 10-mm-diameter puncher, and the weight difference between the two ears was used as the swelling degree.

To determine the influence of the egg-white-induced pedal swelling in the rats, 35 male SD rats (170–220 g) were randomly divided into 7 groups by body weight, with 5 rats per group. All groups were intragastrically administered the microspheres once daily for 3 consecutive days, except the aspirin group, which received the microspheres via intragastric administration on the day of the experiment. The control group received the same volume of 1% CMC-Na (carboxymethyl cellulose-sodium). The gavage volume was 10 mL/kg per group. Thirty minutes after the last gastric gavage, 0.1 mL of fresh egg white was injected subcutaneously into the pedal of the right rear foot of each rat to induce inflammation. The foot areas before inflammation and at 0.5, 1, 2, 3, 4, and 5 h after inflammation were measured. The difference in the foot area before and after inflammation was determined as the degree of swelling, and the swelling rate was calculated.
$$\mathrm{Swell}\;\mathrm{rate}=\frac{A_{t1}-A_{t0}}{A_{t0}\;}\times 100\%$$
A_t0_: foot area before administrationA_t1_: foot area after administration

## 3. Results and Discussion

### 3.1. Characterization of Copolymer mPEG-PLA

Figure 2 shows the nuclear magnetic resonance (NMR) spectra of the mPEG-PLA copolymer. The ^1^H NMR image shows the characteristic peaks of hydrogen in the mPEG-PLA. The peaks at 5.19 ppm and 1.5 ppm correspond to the PLA’s methine and methyl peaks, respectively. The methylene peak at 3.6 ppm is the repeating unit in the PEG. The peak at 3.39 ppm corresponds to the hydrogen of the CH_3_O- group of the mPEG at the end of the copolymer. NMR spectroscopy was performed at 25 °C (Bruker 400 MHz, Karlsruhe, Germany).

Figure 3 shows the mPEG-PLA gel permeation chromatography (GPC) curve with the different molecular weights of mPEG. Data were processed using different molecular weights of polystyrene as the reference material. Table 1 shows the weight-average molecular weight (Mw), number-average molecular weight (Mn) and polydispersity index of the molecular weight distribution (PDI) of the copolymer. The obtained copolymers have relatively small PDIs and relatively uniform molecular weights. Four copolymers with different molecular weights were prepared by copolymerization of four mPEG with the same amount of L-LA. The GPC curves show that the different molecular weights of the four mPEG-PLA, they all have a single peak, indicating that no other copolymers were formed. The copolymer synthesized from mPEG of different molecular weights, wherein the larger the molecular weight of mPEG, the higher the molecular weight of the obtained copolymer and the shorter the peak time of the copolymer, which is consistent with the results obtained by the GPC test.

### 3.2. Morphology and Size Distribution of the Microspheres

Figure 4 shows the scanning electron microscopy (SEM) images of the mPEG-PLA microspheres loaded with ASAs. Figure 4A shows mPEG_550_-PLA microspheres, which have a small particle diameter, a round smooth spherical shape, and no obvious adhesion. Figure 4B shows mPEG_2000_-PLA microspheres with small pores on the surface and varied particle sizes. Figure 4C shows mPEG_5000_-PLA microspheres with a few small pores on the surface and with different pore sizes and numbers. Figure 4D shows mPEG_10000_-PLA microspheres. Compared with the mPEG_550_-PLA, mPEG_2000_-PLA, and mPEG_5000_-PLA microspheres, the mPEG_10000_-PLA’s particle size is larger, its surface has few small pores and its shape is smooth. The adhesion is due to the relatively large molecular weight of the mPEG-PLA copolymer. The pores on the surface are caused by water molecule diffusion during lyophilization. Figure 5 shows the particle size distribution of the mPEG-PLA microspheres loaded with ASAs using a laser particle-size analyzer (Mastersizer 3500, Microtrac, Malvern, UK). The average diameter of the mPEG_550_-PLA microspheres was 1.803 ± 0.21 μm, and the PDI was 0.15. The average diameter of the mPEG_2000_-PLA microspheres was 2.083 ± 0.17 μm, and the PDI was 0.14. The average diameter of the mPEG_5000_-PLA microspheres was 2.631 ± 0.2 μm, and the PDI was 0.20. The average diameter of the mPEG_10000_-PLA microspheres was 3.84 ± 0.30 μm, and the PDI was 0.18(Table 2). These results are nearly the same as those shown in the SEM images. The experimental results showed that the diameters of the 4 microspheres were smaller than 5 μm, indicating that these microspheres are suitable for oral of drug delivery, which is consistent with the study by Wei et al. [34].

### 3.3. Drug Loads

mPEG-PLA microspheres loaded with ASAs were prepared using the W/O/W double-emulsion technique, and the EE and LE were measured by high-performance liquid chromatography (HPLC; Table 2). With the increased molecular weight, the microsphere’s EE and LE increased, with trends similar to those of the mPEG-PLA microspheres loaded with recombinant human growth hormone (rhGH) prepared by Yi et al. [35]. While preparing the microspheres, a portion of the drug diffused into the external water phase due to solvent evaporation, resulting in drug loss. A copolymer with a high molecular weight has a large molecular gap, enabling the drug to diffuse outward, thus increasing drug loss.

### 3.4. Analysis of In Vitro Microsphere Release

Figure 6 shows the release curve of mPEG-PLA microspheres loaded with ASAs in phosphate-buffered saline (PBS) (pH = 7.40). The cumulative amount of mPEG_550_-PLA microspheres released on the first day was 24.32 ± 0.37%, and then, the release rate slowed, reaching 64.38 ± 1.21% after a two-week stable release. The cumulative amount of mPEG_2000_-PLA microspheres released on the first day was 20.74 ± 0.29%, and then, the release rate slowed, reaching 57.09 ± 1.28% after a two-week stable release. The cumulative amount of mPEG_5000_-PLA microspheres released on the first day was 19.54 ± 0.22%, and then, the release rate slowed, reaching 52.45 ± 1.28% after a two-week stable release. The cumulative amount of the mPEG_10000_-PLA microspheres released on the first day was 15.03 ± 0.51%, and then, the release rate slowed, reaching 45.12 ± 1.04% after a two-week stable release. The relatively large amount of the drug released on the first day occurred because a small amount of the drug adhered to the microsphere surface when the microspheres encapsulated the drug. With the microsphere’s increased molecular weight, in vitro release of the drug from the microspheres was relatively slow. The release time of the mPEG-PLA microspheres in this study was longer than that of the microspheres prepared by Zheng et al. [36] and Xiong et al. [37], indicating that the mPEG-PLA was well encapsulated on the ASAs.

### 3.5. Analysis of In Vitro Microsphere Degradation

Based on the morphology, particle size, EE, LE, and in vitro microsphere release, we selected mPEG_10000_-PLA as the optimal microspheres for the study. Figure 7 shows the degradation process of the ASAs-loaded mPEG_10000_-PLA microsphere. The figure shows the changes in the microsphere’s dry weight, system pH and molecular weight over time. In PBS (pH = 7.40, 10 mM), the three curves all decreased with time. Within 60 days, the microspheric pH decreased from 7.40 to 5.89, the microspheres lost 47.16% of their dry weight, and their molecular weight dropped from 10307 Da to 8258 Da. As its degradation progressed, the mPEG-PLA was hydrolyzed to produce CO_2_, which was dissolved in water to decrease the pH. mPEG-PLA is an amphiphilic material that absorbs water in PBS and breaks the hydrophilic segment, thus decreasing the molecular weight. As the hydrophilic segment breaks, the molecular weight of the copolymer decreases, resulting in a loss of copolymer quality. Li et al. [38] studied the degradation of mPEG-PLA nanoparticles and found that the mPEG-PLA degraded very slowly, and the Mn decreased by 27.6% within 30 days. Simone [39] et al. studied the polymer degradation conditions at pH and found that the pH decreased correspondingly over time, which is similar to the mPEG-PLA degradation trend in this study.

### 3.6. Analysis of In Vitro Hemolytic Properties of Microspheres

Figure 8 shows the hemolysis and anticoagulation rates of the ASAs-loaded mPEG_10000_-PLA microsphere, the mPEG_10000_-PLA, and the ASAs. The hemolysis rates of the ASAs-loaded mPEG_10000_-PLA microsphere, the mPEG_10000_-PLA, and the ASAs increased as the concentration increased but did not exceed 5% at 40 μg/mL. Studies have shown that hemolysis rates below 5% indicate blood compatibility, which is consistent with the International Organization for Standardization (ISO) hemolysis standard, indicating applicability for intravenous injection. The anticoagulant chart shows that the blood clotting index (BCI) increases as the concentration increases, indicating that its anticoagulant activity increases as the sample concentration increases.

### 3.7. In Vitro Cytotoxicity Analysis of The Microspheres

Figure 9 shows the cytotoxicity of the ASAs-loaded mPEG_10000_-PLA microspheres, the mPEG_10000_-PLA, and the ASAs to the HL-7702 (normal human hepatocyte) cell line. To study the cytotoxicity of the ASAs-loaded mPEG_10000_-PLA microspheres, the 3-(4,5-dimethylthiazol-2-yl)-2,5-diphenyltetrazolium bromide (MTT) assay was used to detect HL-7702 cell viability under different concentrations of ASAs-loaded mPEG_10000_-PLA microspheres, mPEG_10000_-PLA, and ASAs. Figure 8 shows the toxicity at concentrations from 0.4 μg/mL to 40 μg/mL. The results showed that mPEG_10000_-PLA microspheres could be used for drug loading. The microspheres prepared by Song et al. [40] showed that the HL-7702 cells were dose- and time-dependent, and as the concentration increased, the HL-7702 cell survival rate decreased, which is similar to the HL-7702 cell viability under the ASAs in this study.

### 3.8. Analysis of the Microspheric Anti-Inflammatory Activity

Table 3 shows the effect of the ASAs on xylene-induced auricle swelling in mice. The ASAs-loaded microspheres significantly inhibited the xylene-induced auricle swelling in the mice. The effect of the ASAs-loaded microspheres at 10 mg/kg was comparable to that of the 0.3 g/kg DP, and the efficacies at 20 and 40 mg/kg were stronger than that of 10 mg/kg aspirin. 

Table 4 shows the effect of the ASAs on egg-white-induced pedal swelling in rats. The ASAs-loaded microspheres significantly inhibited the egg-white-induced pedal swelling in the rats, and the action time of the ASAs-loaded microspheres at 20 mg/kg was longer than that of the DP, and the action time of ASAs-loaded microspheres at 80 mg/kg was similar to that of aspirin. 

## 4. Conclusions

ASAs-loaded mPEG-PLA microspheres were successfully prepared using the W/O/W double-emulsion technique. The comprehensive results for the microspheres, including particle size, EE, LE, in vitro release and other tests, showed that mPEG_10000_-PLA microspheres performed excellently. The particle size of ASAs-loaded mPEG_10000_-PLA microspheres was 3.84 ± 0.30 μm, the EE was 75.12 ± 4.25%, and the LE was 5.17 ± 0.19%. The microspheres sustained release for 15 days in a simulated human environment, with beneficial biocompatibility at 40 μg/mL, and showed no toxicity to HL-7702 cells. It significantly inhibited auricle swelling in mice and pedal swelling in rats. These results indicate that mPEG-PLA is a carrier material suitable for controlled ASAs release.

## Figures and Tables

**Figure 1 materials-12-01457-f001:**
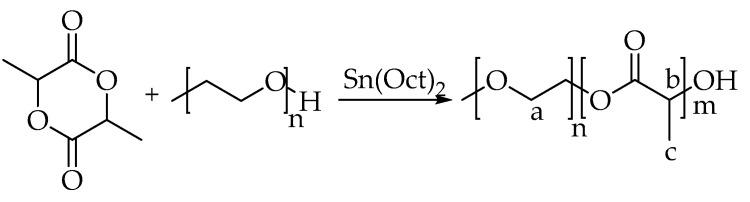
The synthesis process of mPEG-PLA.

**Figure 2 materials-12-01457-f002:**
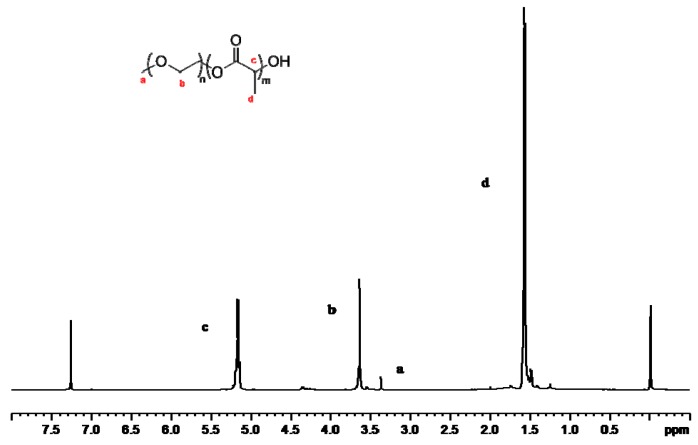
^1^H nuclear magnetic resonance (NMR) spectrum of mPEG-PLA.

**Figure 3 materials-12-01457-f003:**
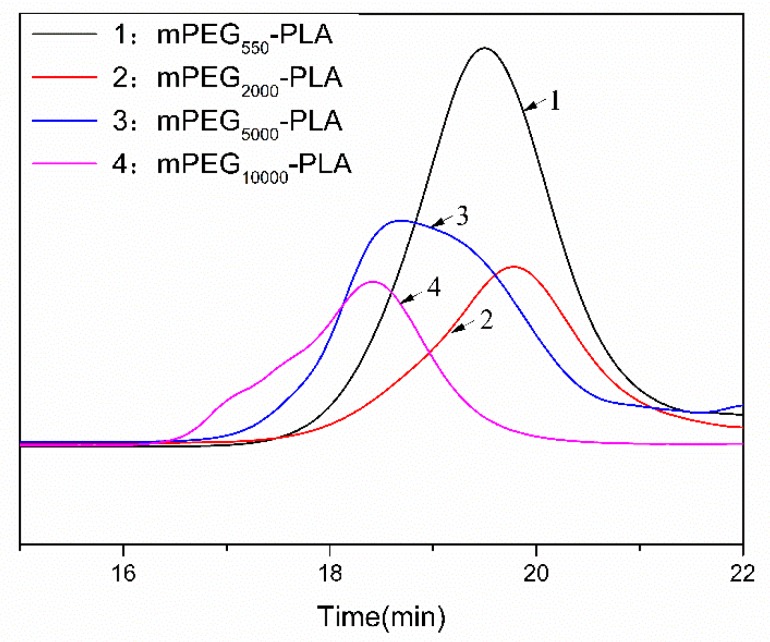
Gel permeation chromatography (GPC) of mPEG-PLA.

**Figure 4 materials-12-01457-f004:**
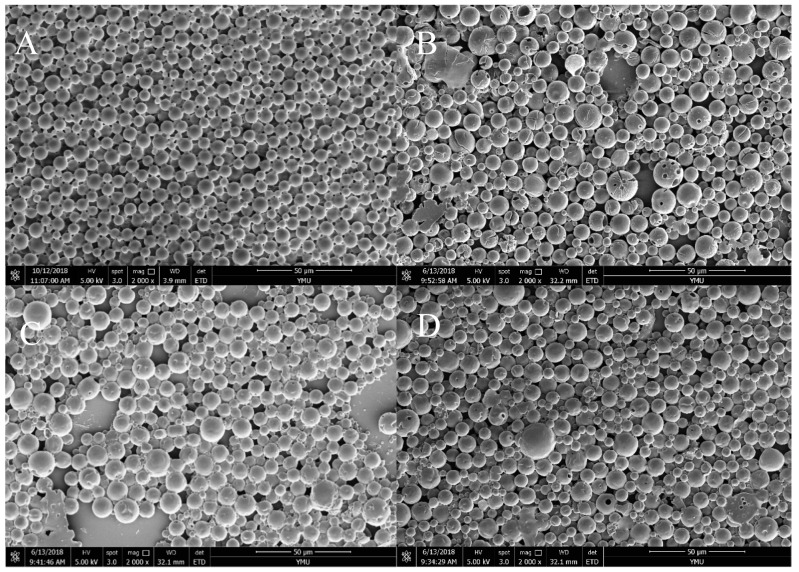
Scanning electron microscopy (SEM) images of mPEG-PLA microsphere. (**A**) mPEG 550-PLA, (**B**) mPEG 2000-PLA, (**C**) mPEG5000-PLA, (**D**) mPEG10000-PLA. The magnification of SEM image was 2000.

**Figure 5 materials-12-01457-f005:**
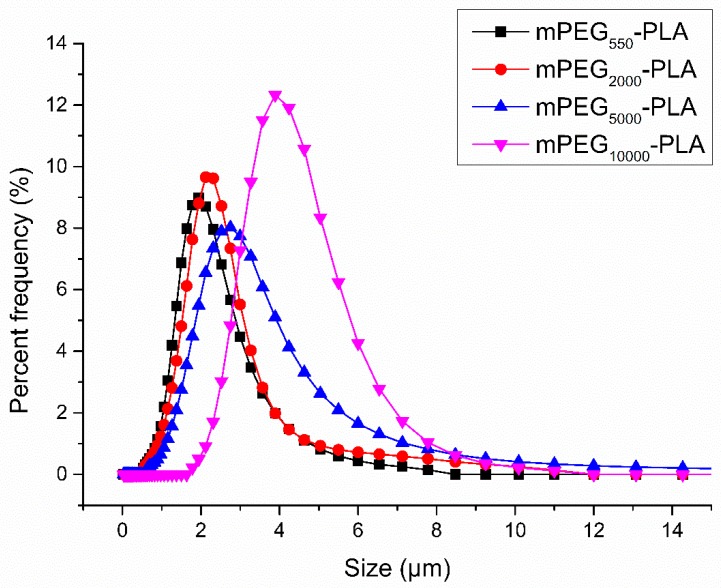
Size distribution of mPEG-PLA microsphere.

**Figure 6 materials-12-01457-f006:**
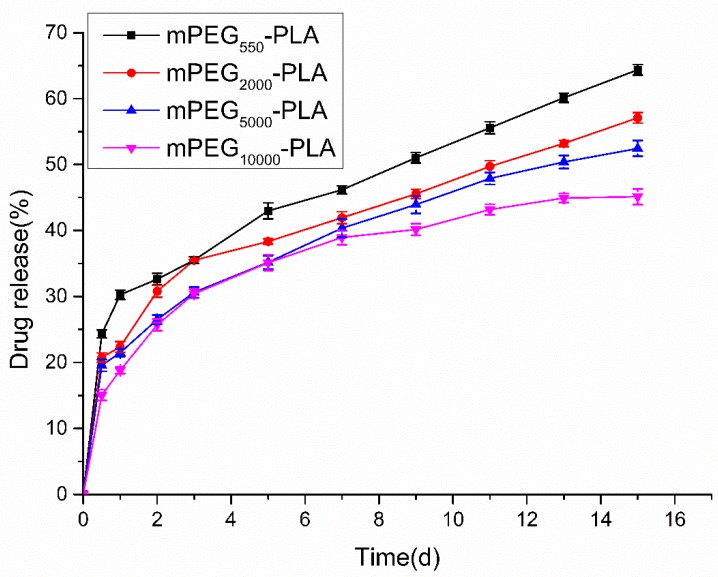
Release curve of mPEG-PLA microspheres loaded ASAs in PBS

**Figure 7 materials-12-01457-f007:**
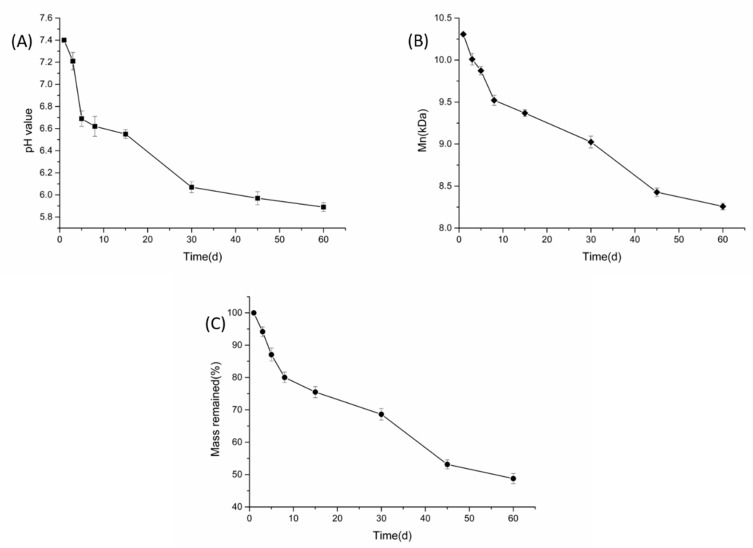
The degradation process of the ASAs-loaded mPEG_10000_-PLA microspheres in the phosphate buffer saline (PBS) (pH = 7.40): the pH (**A**) Mn (**B**) and mass (**C**) changed with time.

**Figure 8 materials-12-01457-f008:**
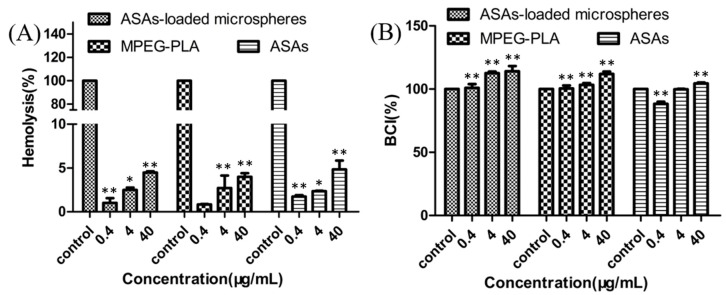
Hemolytic activity (**A**) and blood clotting index (**B**) of the ASAs-loaded mPEG_10000_-PLA microsphere. * *P* < 0.05 and ** *P* < 0.01, compared with control, and the whole blood sample with the ASAs was used as a control.

**Figure 9 materials-12-01457-f009:**
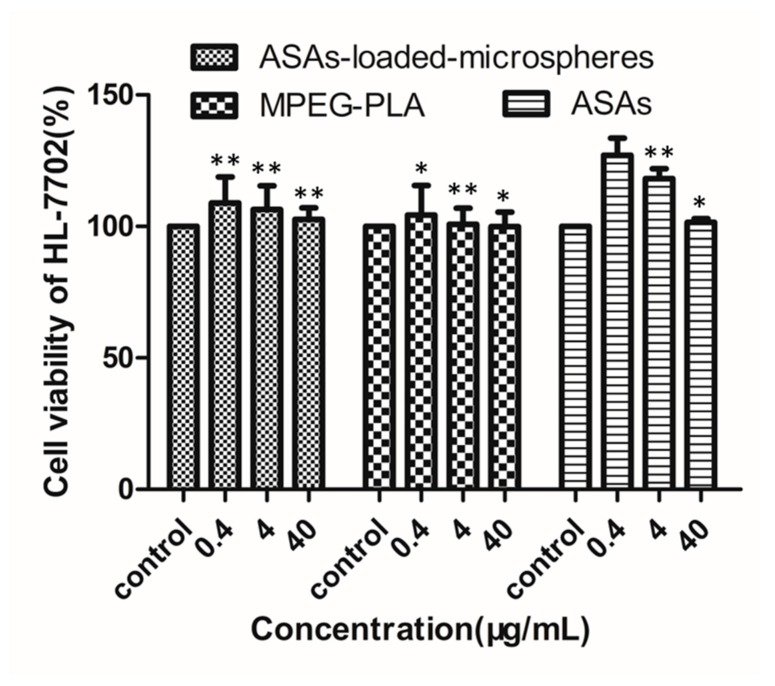
The cytotoxicity of the ASAs-loaded mPEG_10000_-PLA microspheres, the mPEG_10000_-PLA, and the ASAs to the HL-7702 cells. * *P* < 0.05 and ** *P* < 0.01, compared with control, and the whole blood sample with the ASAs was used as a control.

**Table 1 materials-12-01457-t001:** The Mn, Mw and PDI of different molecular weight of mPEG-PLA.

Sample	Mn^1^ (g·mol ^−1^)	Mn^2^ (g·mol ^−1^)	Mw^1^ (g·mol ^−1^)	PDI ^1^
mPEG_550_-PLA	4013	4097	4540	1.13
mPEG_2000_-PLA	5489	5504	6320	1.15
mPEG_5000_-PLA	7378	7419	8637	1.17
mPEG_10000_-PLA	11,307	11,416	12,889	1.14

^1^ Determined by GPC via a universal calibration curve and appropriate Mark−Houwink parameters for THF. ^2^ Calculated by ^1^H NMR. Mn: The number-average molecular weight; Mw: The weight-average molecular weight; PDI: polydispersity index of the molecular weight distribution.

**Table 2 materials-12-01457-t002:** The average diameter, PDI, EE, LE of mPEG-PLA microspheres loaded ASAs.

Sample	Average Diameter (μm)	^1^PDI	^1^EE (%)	^1^LE (%)
mPEG_550_-PLA	1.803 ± 0.21	0.15	53.97 ± 2.57	2.71 ± 0.21
mPEG_2000_-PLA	2.083 ± 0.17	0.14	62.78 ± 3.44	3.05 ± 0.11
mPEG_5000_-PLA	2.631 ± 0.20	0.20	70.28 ± 3.61	4.14 ± 0.12
mPEG_10000_-PLA	3.840 ± 0.30	0.18	75.12 ± 4.25	5.17 ± 0.19

^1^PDI: polydispersity index of the particle size distribution. EE: encapsulation efficiency. LE: loading efficiency.

**Table 3 materials-12-01457-t003:** Effect of the ASAs on xylene-induced auricle swelling in mice.

Group	Doses (/kg)	Auricular Swelling Degree ($\bar{x}$ ± s, mg)	Inhibition Rate (%)
Control	—	50.85 ± 3.61	—
Aspirin	10 mg	34.65 ± 2.90	31.86%
DP	0.3 g	37.65 ± 3.04	25.96%
ASAs	0.1 g	39.65 ± 0.78	22.03%
	10 mg	39.00 ± 2.69	23.30%
Loaded	20 mg	31.35 ± 6.86	38.35%
microspheres	40 mg	27.65 ± 7.57	45.62%

**Table 4 materials-12-01457-t004:** Effect of the ASAs on egg-white-induced pedal swelling in rats.

Group	Doses (/kg)	Inhibitory Swelling Rate of Sole (%)				
		1h	2h	3h	4h	5h
Control	—	—	—	—	—	—
Aspirin	20 mg	66.77	62.07	57.02	52.17	49.55
DP	0.24 g	68.81	65.86	60.39	57.31	54.48
ASAs	0.16 g	78.97	69.99	68.29	60.14	57.87
Loaded microspheres	20 mg	51.73	47.12	43.72	38.97	35.14
	40 mg	59.62	55.47	46.64	44.61	41.78
	80 mg	75.37	73.22	65.14	58.07	55.27

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
