# Peer review of "Preparation and In Vitro Release of Total Alkaloids from *Alstonia Scholaris* Leaves Loaded mPEG-PLA Microspheres"

_materials, 2019, doi:10.3390/ma12091457_

Reviewer 1 Report

Authors used standard PEG-b-(L-PLA) copolymer and standard W/O/W method for preparation of particles loaded with alkaloids extracted from Alstonia scholaris leaves. Thus, the work is not novel (there are already a few hundred of papers on similar carriers), however, results of the studies are of significant practical importance for pharmaceutical  applications. General plan of experiments was properly designed and results were analyzed using appropriate methods. However, presentation should be significantly improved. In current state the paper is often confusing, what reading tiring. From the beginning of the Materials subsection the reader may think that authors used L-lactic acid as a substrate for copolymer synthesis (they wrote that they purchased this compound). However, later they claim that the synthesized copolymer was obtained from L-lactide by ring-opening polymerization. This point must be clarified. In page 3, line 92 authors wrote that ASAs were dissolved in methanol solution (solution containing what?). I guess that they dissolved ASAs in methanol solvent. Molar masses of copolymers were determined by GPC. However, nothing is written about equipment and raw data treatment. Authors wrote that Figures 3A, 3B and 3C show SEM microphotographs of particles, whereas the SEM microphotographs are shown in Figure 4. Figure 5 displays the particle diameter distribution. However, authors did not mention whether plots were prepared on the basis of analysis of SEM microphotographs or by light scattering. In the latter case authors should mention the upper limit of particle diameters, which could be measured using equipment they had in their disposal. Authors did not discuss one of the most important aspect of their work, namely, route of drug application (oral, pulmonary, intravenous?). This is crucial, because depending on the route of delivery the particles with appropriate particle diameters should be prepares and relevant biological tests should be selected. 

Author Response

Response to Reviewer 1 Comments

Ms. Ref. No. ID: materials-488518, Title: Preparation and in vitro release of mPEG-PLA microspheres loaded with total alkaloids from Alstonia scholaris leaves. Journal of materials

Dear editor,

I have made revisions according to reviewers’ comments. It had been appended below. I hope you can think about my paper again.

Yours sincerely,

Minglong Yuan

From the beginning of the Materials subsection the reader may think that authors used L-lactic acid as a substrate for copolymer synthesis (they wrote that they purchased this compound). However, later they claim that the synthesized copolymer was obtained from L-lactide by ring-opening polymerization. This point must be clarified.

Response: Thanks for your kind suggestion. The synthesized copolymer was obtained from L-lactide by ring-opening polymerization. L-lactide was purchased from Purac Co., Ltd. (Shanghai, China). The description is worry, and I have corrected it. See line 66.

In page 3, line 92 authors wrote that ASAs were dissolved in methanol solution (solution containing what?). I guess that they dissolved ASAs in methanol solvent.

Response: I have corrected it. The description is not clear. The ASAs were dissolved in methanol solvent. See line 104-105.

Molar masses of copolymers were determined by GPC. However, nothing is written about equipment and raw data treatment.

Response:

2.3 Determination of molecular mass

3 mg copolymer were dissolved in 1 mL chromatography grade THF and was filtered by using a 0.45 μm nylon 66 filter membrane. Then using gel permeation chromatography (GPC) was purchased from Waters Inc. (Milford, MA) to define the molecular weight and distribution of the compounds, and THF was used as the eluent. The system is equipped with a column (7.8 * 300 mm, Waters Styragel), a Waters 515 pump and a Waters 2414 refractive index detector. When the column temperature of GPC is 40 , the flow rate is 1 mL/min, and the baseline is smooth. The filtrate was pulled into an injection needle, and the sample was slowly and uniformly injected into the sampler when the air was removed. All data were obtained under the same standard curve. See line 93-101.

Authors wrote that Figures 3A, 3B and 3C show SEM microphotographs of particles, whereas the SEM microphotographs are shown in Figure 4. Figure 5 displays the particle diameter distribution. However, authors did not mention whether plots were prepared on the basis of analysis of SEM microphotographs or by light scattering. In the latter case authors should mention the upper limit of particle diameters, which could be measured using equipment they had in their disposal.

Response: I have corrected Figure 3A, 3B, 3C and 3D to 4A, 4B and 4C. Figure 5 was prepared by a laser particle-size analyzer (Mastersizer 3500, Microtrac, UK). Particle size: 0.01 - 3500 µm. See line 268-270, 272,277-278.

Authors did not discuss one of the most important aspect of their work, namely, route of drug application (oral, pulmonary, intravenous?). This is crucial, because depending on the route of delivery the particles with appropriate particle diameters should be prepares and relevant biological tests should be selected.

Response: The experimental results showed that the diameters of the 4 microspheres were smaller than 5 μm, indicating that these microspheres are suitable for oral of drug delivery. See line 284.

Reviewer 2 Report

The authors present an interesting study focused on the use of MPEG-PLA microspheres as carriers of alcaloids derived from Astonia scholaris. Despite the potential interest of the research, in general, the manuscript is difficult to follow and should be improved in several aspects.  

Title. If the microspheres are aimed for the transport of alkaloids, the title doesn’t summarize this fact. The reader understands that mPEG-PLA microspheres are released instead of the alkaloids.

Abstract. The abstract should also be rewritten. Good biological activity is a too broad concept. Do the microspheres really encapsulate the total alkali? The sentence from line 20 to line 24 is too long.

Introduction.  The wording of the first paragraph should be revised. Line 38-39, …..effectively, treats respiratory… The alkaloid doesn’t treat but is used to treat.

Methods. Line 105, What is the meaning of… but the internal water phase volume was the same of the pure water?

Line 111….were observed on the observation area. It is obvious.

Lines 127-128. The sentence is not understandable.

Equations should be numbered. Parameters of the equation of line 133-134, do not match with the information of lines 137 and 138.

Line 188, d should be replaced by days

Results. In general, figure legends are too poor and should give more information.

Line 237. The authors state that the synthesized copolymers are same and no other copolymers were formed. What do they mean? If one looks at figure 3, the samples are very polydisperse. Table 1 shows several parameters of the copolymers but they are poorly described. Mn is manganese for the reader, MW refers to molecular weight? PDI is the polydispersity index? If so, this value goes from 0 to 1. All the values should be given with the corresponding standard deviation.

Lines 256- 265. Sizes should be summarized in a table.

Author Response

Response to Reviewer 2 Comments

Ms. Ref. No. ID: materials-488518, Title: Preparation and in vitro release of mPEG-PLA microspheres loaded with total alkaloids from Alstonia scholaris leaves. Journal of materials

Dear editor,

I have made revisions according to reviewers’ comments. It had been appended below. I hope you can think about my paper again.

Yours sincerely,

Minglong Yuan

The authors present an interesting study focused on the use of MPEG-PLA microspheres as carriers of alcaloids derived from Astonia scholaris. Despite the potential interest of the research, in general, the manuscript is difficult to follow and should be improved in several aspects.

Title. If the microspheres are aimed for the transport of alkaloids, the title doesn’t summarize this fact. The reader understands that mPEG-PLA microspheres are released instead of the alkaloids.

Response: Thanks for your kind suggestion. I have corrected the title to Preparation and in vitro release of the total alkaloids from Alstonia scholaris leaves loaded the mPEG-PLA microspheres. See line 2-5.

Abstract. The abstract should also be rewritten. Good biological activity is a too broad concept. Do the microspheres really encapsulate the total alkali?

Response: I have corrected the ’good biological activity’ to the ’positive anti-inflammatory activity’. The total alkaloids from Alstonia scholaris leaves contain two main alkaloids in our study. In the drug release process of microspheres, the drug was released by crushing the encapsulated material, and the release solution was detected by HPLC. Two main alkaloids peaks were detected, so it was considered that the purpose of encapsulating drugs was achieved. See line 18.

The sentence from line 20 to line 24 is too long.

Response: I have corrected it. The ASAs-loaded mPEG10000-PLA microspheres were screened for better performance by testing the morphology, average particle size, embedding rate and drug loading of different molecular weight mPEG-PLA microspheres, which can stably and continuously release for 15 days at 37 °C. The results of cytotoxicity and blood compatibility indicated that the drug-loaded microspheres have beneficial biocompatibility. See line 21-26.

Introduction. The wording of the first paragraph should be revised. Line 38-39, …..effectively, treats respiratory… The alkaloid doesn’t treat but is used to treat.

Response: I have corrected as beneficial effects respiratory. See line 40.

Methods. Line 105, What is the meaning of… but the internal water phase volume was the same of the pure water?

Response: I have corrected as ’Blank microspheres were prepared by the same method’. See line116-117.

Line 111….were observed on the observation area. It is obvious.

Response: I have corrected it. A few lyophilized microspheres (white powder) were evenly spread on a conductive adhesive, and the surface was sprayed with gold to observe. See line 123.

Lines 127-128. The sentence is not understandable.

Response: The sentence means the supernatant were collected periodically and replaced with fresh buffer of equal volume.

Equations should be numbered. Parameters of the equation of line 133-134, do not match with the information of lines 137 and 138.

Response: I’m so sorry that I miswrote it, and I have corrected Vt to V1, Vs to V2. See line150-151.

Line 188, d should be replaced by days

Response: I have corrected d to days. See line 201.

Results. In general, figure legends are too poor and should give more information.

Line 237. The authors state that the synthesized copolymers are same and no other copolymers were formed. What do they mean? If one looks at figure 3, the samples are very polydisperse. Table 1 shows several parameters of the copolymers but they are poorly described. Mn is manganese for the reader, MW refers to molecular weight? PDI is the polydispersity index? If so, this value goes from 0 to 1. All the values should be given with the corresponding standard deviation.

Lines 256- 265. Sizes should be summarized in a table.

Response: Four copolymers with different molecular weights were prepared by copolymerization of four mPEG with the same amount of L-LA. The GPC curves show that the different molecular weights of the four mPEG-PLA, they all have a single peak, indicating that no other copolymers were formed.

According to the definition of polymers PDI, that is used to describe the molecular weight distribution of polymers. PDI was calculated by Mw/Mn, and the value of Mw is larger than that of Mn, so the PDI value obtained can hardly be less than 1. See line 248-251, Table 1 and 2.

Table 1. The Mn, Mw and PDI of different molecular weight of mPEG-PLA

Sample

Mn1

(g·mol-1)

Mn2

(g·mol-1)

Mw1

(g·mol-1)

PDI1

mPEG550-PLA

4013

4097

4540

1.13

mPEG2000-PLA

5489

5504

6320

1.15

mPEG5000-PLA

7378

7419

8637

1.17

mPEG10000-PLA

11307

11416

12889

1.14

1 Determined by GPC via a universal calibration curve and appropriate Mark−Houwink parameters for THF.

2 Calculated by 1H NMR.

Mn: The number-average molecular weight; Mw: The weight-average molecular weight; PDI: polydispersity index of the molecular weight distribution.

Table 2. The average diameter, PDI, EE, LE of mPEG-PLA microspheres loaded ASAs

Sample

average diameter   (μm)

PDI

EE (%)

LE(%)

mPEG550-PLA

1.803±0.21

0.15

53.97±2.57

2.71±0.21

mPEG2000-PLA

2.083±0.17

0.14

62.78±3.44

3.05±0.11

mPEG5000-PLA

2.631±0.20

0.20

70.28±3.61

4.14±0.12

mPEG10000-PLA

3.840±0.30

0.18

75.12±4.25

5.17±0.19

Reviewer 3 Report

Please, find below some points for revision.

The authors refer through the manuscript to ‘good effect on’, ‘good drug carriers’. I would recommend to replace ‘good’ with a positive, efficient, beneficial …

Page 1 line 37: Please delete ‘from the Kunming Institute of 37 Botany, Chinese Academy of Sciences’.

The method for GPC analysis should be described in the experimental section.

Page 6: the authors discuss the GPC analysis of mPEG-PLA copolymers. According to text, the molecular weight of the copolymers was determined using polystyrene as reference material. However, the hydrodynamic volume of polystyrene is different compared to the hydrodynamic volume of block copolymers. Therefore, to have an accurate estimate of the molecular weight at least two detectors should be used.

Figure 6: error bars should be provided.

Figure 8 and 9: a statistical analysis should be provided. If the differences are not significant this should be stated in the text.

Author Response

Response to Reviewer 3 Comments

Ms. Ref. No. ID: materials-488518, Title: Preparation and in vitro release of mPEG-PLA microspheres loaded with total alkaloids from Alstonia scholaris leaves. Journal of materials

Dear editor,

I have made revisions according to reviewers’ comments. It had been appended below. I hope you can think about my paper again.

Yours sincerely,

Minglong Yuan

The authors refer through the manuscript to ‘good effect on’, ‘good drug carriers’. I would recommend to replace ‘good’ with a positive, efficient, beneficial.

Response: Thanks for your kind suggestion. I would replace ‘good effect on’ with ’positive effect on’, ‘good drug carriers’ with ‘efficient drug carriers’, ’good biological activity’ with ’positive biological activity’, ’good biocompatibility’ with ’beneficial biocompatibility’, ’good anti-inflammatory’ with ’beneficial anti-inflammatory’. See line 18, 25-26, 40, 388.

Page 1 line 37: Please delete ‘from the Kunming Institute of 37 Botany, Chinese Academy of Sciences’.

Response: I have corrected it. See line 38-39.

The method for GPC analysis should be described in the experimental section.

Response:

2.3 Determination of molecular mass

3 mg copolymer were dissolved in 1 mL chromatography grade THF and was filtered by using a 0.45 μm nylon 66 filter membrane. Then using gel permeation chromatography (GPC) was purchased from Waters Inc. (Milford, MA) to define the molecular weight and distribution of the compounds, and THF was used as the eluent. The system is equipped with a column (7.8 * 300 mm, Waters Styragel), a Waters 515 pump and a Waters 2414 refractive index detector. When the column temperature of GPC is 40 , the flow rate is 1 mL/min, and the baseline is smooth. The filtrate was pulled into an injection needle, and the sample was slowly and uniformly injected into the sampler when the air was removed. All data were obtained under the same standard curve. See line 93-101.

Page 6: the authors discuss the GPC analysis of mPEG-PLA copolymers. According to text, the molecular weight of the copolymers was determined using polystyrene as reference material. However, the hydrodynamic volume of polystyrene is different compared to the hydrodynamic volume of block copolymers. Therefore, to have an accurate estimate of the molecular weight at least two detectors should be used.

Response: At present, the molecular weight of copolymers is mainly used polystyrene as reference material. Meanwhile, the Mn data calculated by NMR identical with the GPC. See Table 1.

Table 1. The Mn, Mw and PDI of different molecular weight of mPEG-PLA

Sample

Mn1

(g·mol-1)

Mn2

(g·mol-1)

Mw1

(g·mol-1)

PDI1

mPEG550-PLA

4013

4097

4540

1.13

mPEG2000-PLA

5489

5504

6320

1.15

mPEG5000-PLA

7378

7419

8637

1.17

mPEG10000-PLA

11307

11416

12889

1.14

1 Determined by GPC via a universal calibration curve and appropriate Mark−Houwink parameters for THF.

2 Calculated by 1H NMR.

Mn: The number-average molecular weight; Mw: The weight-average molecular weight; PDI: polydispersity index of the molecular weight distribution.

Figure 6: error bars should be provided.

Response: I have corrected the error bars. See Figure 6.

Figure 8 and 9: a statistical analysis should be provided. If the differences are not significant this should be stated in the text.

Response: I have corrected a statistical analysis. See Figure 8 and 9.

Round  2

Reviewer 1 Report

The revised version could be accepted for publication.

Author Response

Thanks for your kind suggestion. We will make persistent efforts in the later work.

Reviewer 2 Report

The authors have made the corrections proposed by the reviewer and the manuscript is suitable for publication.

Author Response

(The authors gave the same response as above.)
